# Prevalence of sleep apnea in children and adolescents in Colombia according to the national health registry 2017–2021

**Alan Waich**[1][☯]*, **Juanita Ruiz Severiche**[1][☯], **Margarita Manrique Andrade**[2][‡], **Julieth Andrea Castañeda Aza**[2][‡], **Julio Cesar Castellanos Ramírez**[3][‡], **Liliana Otero Mendoza**[1,4][‡], **Sonia Maria Restrepo Gualteros**[1,5,6][‡], **Olga Patricia Panqueva**[1,5,6][‡], **Patricia Hidalgo Martínez**[1,5,7][‡]

1 Sleep disorders research group, School of Medicine, Pontificia Universidad Javeriana, Bogotá, Colombia, 2 Research Office, Hospital Universitario San Ignacio, Bogotá, Colombia, 3 General direction, Hospital Universitario San Ignacio, Bogotá, Colombia, 4 Center of Dental Research, Pontificia Universidad Javeriana, Bogotá, Colombia, 5 Sleep clinic, Hospital Universitario San Ignacio, Bogotá, Colombia, 6 Departament of Pediatrics, Hospital Universitario San Ignacio, Pontificia Universidad Javeriana, Bogotá, Colombia, 7 Departament of Internal Medicine, Hospital Universitario San Ignacio, Pontificia Universidad Javeriana, Bogotá, Colombia

☯ These authors contributed equally to this work.
‡ These authors also contributed equally to this work
* waichalan@javeriana.edu.co

**Data Availability Statement:** All relevant data are within the paper and its Supporting Information file.

## Abstract

### Objective

To describe the sociodemographic and epidemiological characteristics of diagnosis and treatment of pediatric patients with sleep apnea, both central and obstructive, in Colombia between 2017 and 2021.

### Methods

Observational, descriptive, cross-sectional, epidemiological study using the International Classification of Diseases and Related Health Problems as search terms for sleep apnea, based on SISPRO, the Colombian national health registry. Stratification by gender and age groups was performed. We also generated data of the amount of diagnostic and therapeutic procedures performed. A map of prevalence by place of residency was performed.

### Results

National records report 15200 cases of SA between 2017 and 2021, for an estimated prevalence of 21.1 cases by 100000 inhabitants in 2019 the year with the most cases (4769), being more frequent and in the 6 to 11 age group and in males, with a male to female ratio of 1.54:1. The number of cases declined in 2020 and 2021. The map showed a concentration of cases in the more developed departments of the country.

**Funding:** Initial of the authors: L.O. Grant Number: 120380763680 Full name of funder: Ministerio Ciencia Tecnología e Innovación de Colombia-Minciencias URL: https://minciencias.gov.co The funders had no role in study design, data collection and analysis, decision to publish, or preparation of the manuscript.

**Competing interests:** The authors have declared that no competing interests exist.

## Discussion

This is the first approximation to a nation-wide prevalence of sleep apnea in Colombia which is lower to what is found in the literature worldwide, including studies performed in Latin America and in Colombia, this could reflect sub diagnosis and sub report. The fact that the highest prevalence was found in males and in the 6–11 age group is consistent with reports in literature. The decrease in cases in 2020 and 2021 could be related to the COVID-19 pandemic impact in sleep medicine services.

## Introduction

Sleep apnea (SA) comprises two main clinical and polysomnographic categories: Central Sleep Apnea (CSA) and Obstructive Sleep Apnea (OSA) [1]. OSA is a respiratory sleep disorder characterized by a partial or total obstruction of the airway associated with altered sleep architecture and intermittent hypoxia [1]. In children and adolescents, a prevalence of 1 to 4% has been reported in multiple studies [2]. Some of the risk factors for OSA in this age groups are hypertrophy of the tonsils, craniofacial abnormalities, and obesity [3,4]. Untreated OSA has important cardiovascular, metabolic, and neurocognitive consequences [5–8]. Obstructive sleep disordered breathing (oSDB) comprises OSA and other problems in which the upper airway is compromised [1]. There are multiple management options for children with oSDB that range from watchful waiting to surgical removal of the tonsils or positive airway pressure devices [9,10]. Evidence of the effect of these therapeutic options on important aspects of disease such as severity, quality of life, polysomnographic values, behavioral and cognitive outcomes have been studied [11,12].

CSA happens when there is diminished, or absent respiratory effort often related to desaturation, nocturnal awakenings, and sleep fragmentation [1]. It has specific polysomnographic criteria [13]. In children and adolescents, CSA is frequently associated with other conditions such as genetic syndromes, laryngomalacia, prematurity, obesity, Arnold-Chiari malformation, among others [14,15]. CSA and OSA can occur concomitantly [16].

In Latin America, the information of the prevalence of SA in children is limited, most of the frequencies described come from questionnaire-based population studies regarding sleep disordered breathing in a specific city of a particular country. This has many limitations: these screening tools are limited in specificity, they depend on parental report which can both over and under-estimate sleep disorders, they do not replace formal clinical and polysomnographic testing [17], and each study has used different screening tools which hinders adequate comparisons.

Healthcare registries are essential to follow the local epidemiology and offer opportune diagnosis to patients [18–20]. The Integrated Social Protection Information System (SISPRO) is a set of databases developed by the Colombian Ministry of Health that collects, and storages information generated by the health and social security system [21] which is very close to universal coverage (95.23% according to the most recent figure) [22]. Through different information sources, data is obtained derived from the Individual Registry of Provision of Health Services (RIPS, for its Spanish acronym), which is mandatory for health personnel to fill out during every inpatient or outpatient medical attention this includes diagnosis using the 10th revision of the International Statistical Classification of Diseases and Related Health Problems (ICD-10). Data uploaded to SISPRO is subject to rigorous quality control [21].

The aim of this study was to describe the sociodemographic and epidemiological characteristics of diagnosis and treatment of pediatric patients with sleep apnea, both central and obstructive, in Colombia between 2017 and 2021.

## Methods

We conducted an observational, descriptive, cross-sectional study. The data of the population with a principal diagnosis of Sleep Apnea (ICD 10 code G47.3: Sleep Apnea: Central and Obstructive), publicly available in RIPS. We obtained information for the whole country, in the period from January 1st, 2017, and December 31st, 2021.

The principal diagnosis (Sleep Apnea) corresponds to the disease that caused the signs and symptoms for which the patient consults to the health institution or independent health provider. It is important to clarify that the SISPRO database does not allow for the differentiation between OSA and CSA since the ICD-10 code G47.3, Sleep Apnea, includes both OSA and CSA.

First consults seen in the period and previously confirmed cases were analyzed. Patients aged 0 to 17 years, male and female, were included and stratified by developmental age groups (less than 1 year, 1–5, 6–11, 12–18). Cases of diagnostic impression were excluded since they are not considered confirmed SA cases. SISPRO contains information of the patients that required a consult for a principal diagnosis of SA trough the Colombian Healthcare System. Variables of interest such as department of residence and attention, type of insurance, number of new and repeated cases, use of diagnostic and therapeutic procedures (surgical and non-surgical) were described.

A descriptive analysis was developed, calculating proportions for qualitative variables and measures of central tendency for quantitative variables. Contingency tables were used for bivariate descriptive analysis. Prevalence and incidence rates were calculated using as denominators the projected and retro projected population of the 2018 Colombian national census [23]. Statistical analysis was performed using R studio (version 4.1.2, 2018). And the software QGIS (2009) was used to design a prevalence map.

This approach is similar to the ones proposed in other studies that have used data from SISPRO to estimate the prevalence of other diseases in Colombia [24–30].

Ethical considerations and data availability:

The study protocol was reviewed and approved by the Research and Ethics Committee of Hospital Universitario San Ignacio and Pontificia Universidad Javeriana, both located in Bogota, Colombia. (FM-CIE-0473-21). The study was classified as no risk research and conducted in agreement with the Helsinki Declaration and Resolution 008430 of 1993 issued by the Colombian Ministry of Health. Data collected for analysis came from SISPRO, the Colombian national health registry [21]. Data is fully anonymized in the source, before being accessed by researchers. Thus, a waiver for informed consent was obtained. The raw data is available publicly or under request at https://www.sispro.gov.co.

## Results

During the five years evaluated, we identified a total of 15.099 registries and a total of 9737 children and adolescents in SISPRO with a principal diagnosis of SA; 5463 children were newly diagnosed with SA in this period. 2019 was the year with the most cases (2996 new and repeated cases) (Table 1 and Fig 1). For all age groups, the highest prevalence was found in 2019 which was also the year with the most cases (4769 cases), this would provide the most precise prevalence estimation of our data of 21.1 cases by 100000 inhabitants in children and adolescents. 2017 was the year with the lowest prevalence (8.8 by 100000 inhabitants)

**Table 1. New and repeated cases of sleep apnea in children and adolescents, Colombia, 2017–2021.**

| Cases / Year | 2017 | 2018 | 2019 | 2020 | 2021 | Total |
|---|---|---|---|---|---|---|
| New cases | 843 | 965 | 1.773 | 1.045 | 837 | **5463** |
| Repeated cases | 395 | 1.029 | 1.223 | 844 | 783 | **4274** |
| **Total** | **1.238** | **1.994** | **2.996** | **1.889** | **1.620** | **9737** |

(Table 2). There was a steady increase of cases reported between 2017 and 2019 and a significant decrease in 2020 and 2021 (Table 1 and Fig 1). Newly diagnosed cases were lowest in 2021 (837) and highest in 2019 (1773).

Regarding age groups, SA was more frequent in children between 6 and 11 years of age in all of the years evaluated. The lowest prevalence was consistently found in the less than 1 year old group, probably because primary apneas of the newborn have a different code in the ICD-10. During the five years of observation, 490 cases of other apneas of the newborn (ICD-10 P28.4) and 81 cases of primary sleep apneas of the newborn (ICD-10 P28.3).

In the 1–5 and in the 6–11 years group the lowest prevalence was found in 2017, and in the 12–17 group the lowest prevalence was found in 2021. (Tables 1 and 2). Prevalence was greater in males, with a male to female ratio of 1.54: 1 in the five year period. The group with the highest prevalence during the time of study were the 6 to 11 years old males in 2019 (40.4 per 100000 inhabitants).

The analysis of the type of insurance of patients with SA (Table 3) showed that the majority belonged to the contributory regime (72.9%), followed by the subsidized regime (21.3%), special regimes (3%), and complementary plans (0.7%).

A prevalence by department of residence analysis was performed for 2019 (Fig 2). We found an ample range of frequencies varying from 0 cases in San Andrés, Amazonas, Guainia and Guaviare to 243.5 per 100000 in Risaralda and 54.6 per 100000 in Bogota, the country's capital city. Remarkably, the prevalence in Risaralda (the department with the highest prevalence) was 4.5 times the one of Bogotá (the second highest prevalence) this is related to population density which is 8 times higher in Bogotá than in Risaralda [23].

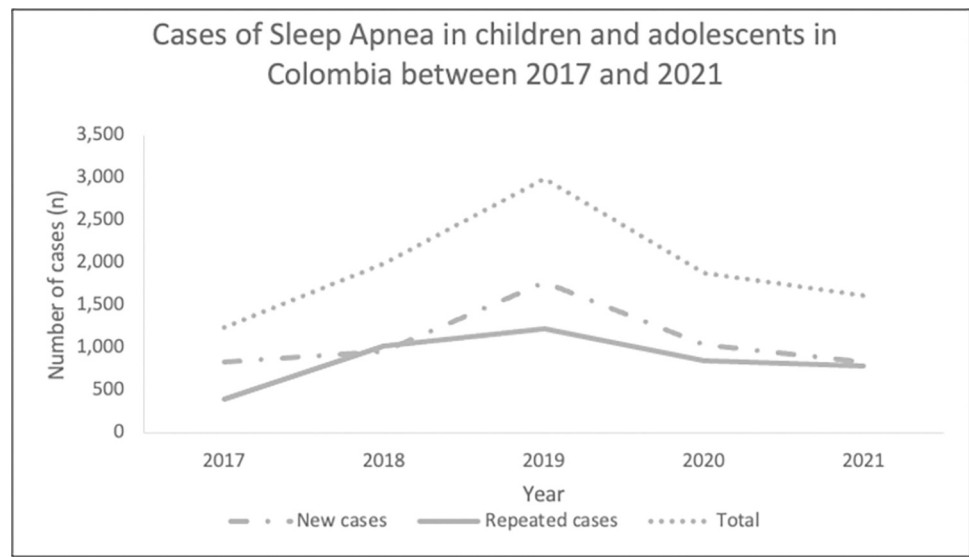

**Fig 1. Cases of sleep apnea in children and adolescents in Colombia between 2017 and 2021.**

**Table 2. Prevalence of sleep apnea in children and adolescents, Colombia, 2017–2021.**

| Year | 2017 | | | 2018 | | | 2019 | | | 2020 | | | 2021 | | | Total | | |
|---|---|---|---|---|---|---|---|---|---|---|---|---|---|---|---|---|---|---|
| Age (years) | Female | Male | Total | Female | Male | Total | Female | Male | Total | Female | Male | Total | Female | Male | Total | Female | Male | Total |
| < 1 | - | - | - | - | - | - | - | - | - | - | - | - | 2.9 | 2.0 | 2.5 | 2,9 | 2,0 | 2,5 |
| 1 to 5 | 3.3 | 3.8 | 3.5 | 5.7 | 6.5 | 6.1 | 11.4 | 17.6 | 14.6 | 10.0 | 15.0 | 12.6 | 10.3 | 13.9 | 12.1 | 40,7 | 56,9 | 49,0 |
| 6 to 11 | 13.0 | 16.3 | 14.7 | 20.1 | 29.6 | 24.9 | 27.9 | 40.4 | 34.3 | 15.3 | 22.1 | 18.8 | 11.9 | 17.8 | 14.9 | 88,1 | 126,1 | 107,5 |
| 12 to 17 | 7.3 | 10.2 | 8.7 | 9.7 | 15.2 | 12.5 | 12.1 | 21.6 | 16.9 | 7.0 | 13.7 | 10.4 | 7.1 | 10.1 | 8.6 | 43,1 | 70,7 | 57,1 |
| Total | 7.7 | 9.9 | 8.8 | 11.5 | 16.8 | 14.2 | 16.5 | 25.5 | 21.1 | 10.2 | 16.1 | 13.2 | 9.3 | 13.2 | 11.3 | 55,1 | 81,5 | 68,6 |

Tables 4 and 5 show the SA related diagnostic and therapeutic procedures to which children and adolescents in SISPRO had performed. 2019 was the year with the highest number of diagnostic procedures executed. In total, 2845 Polisomnographies, 449 capnographies 1225 nasolaringoscopies and 213 transthoracic echocardiographies were carried out in the five years. Only polysomnography is regarded as a procedure for SA diagnosis, the other three procedures (Capnography, Nasolaringoscopy and transthoracic echocardiography) can aid the diagnosis and/or management of pediatric SA patients. Regarding therapeutic procedures, 882 and 855 amigdalectomies and adenoidectomies were performed, respectively. Only 41 patients had positive airway pressure therapy (PAP) initiated.

## Discussion

This is the first approximation to a nation-wide prevalence of SA analyzing data from all the regions of Colombia. When comparing the estimated prevalence of SA in children and adolescents in our study (21.1 by 100000 inhabitants in 2019) is lower than the ones reported in the aforementioned questionnaire-based studies worldwide [2,17,31].

Since most prevalence population studies have been performed for sleep disordered breathing and not a formal diagnosis of SA, comparisons are limited with previous literature. In general, these studies have a higher sensitivity but lower specificity when compared with the analysis performed from health registries such as SISPRO. Nevertheless, this prevalences do appear to be higher than our study. This could reflect sub diagnosis and sub report of SA in Colombia, since the patients from the questionnaire studies might have not consulted and given a formal diagnosis within the Colombian Healthcare system and therefore, are not part of the SISPRO data.

We must highlight that healthcare coverage in Colombia is almost universal (95.23%) [22], and SISPRO is a strong healthcare registry with constant quality controls [21], this is important since it allows us to infer that there are few patients with SA as a principal diagnosis that are not being counted in the study. Our findings reinforce the necessity of strengthening SA early detection, appropriate evaluation, and management in Colombia. Since the patients in our study were given a principal diagnosis of SA by a healthcare professional, they should have

**Table 3. Type of insurance of children and adolescents with sleep apnea, Colombia, 2017–2021.**

| Insurance | 2017 | 2018 | 2019 | 2020 | 2021 | Total |
|---|---|---|---|---|---|---|
| Contibutory | 678 | 741 | 1.296 | 732 | 601 | 3.700 |
| Subsidized | 156 | 159 | 363 | 242 | 233 | 1083 |
| Special | 3 | 50 | 52 | 31 | 30 | 150 |
| Not applicable / uninsured | 5 | 14 | 61 | 22 | 31 | 132 |
| Complementary plans | 2 | 0 | 1 | 17 | 15 | 34 |
| **Total** | **839** | **962** | **1.766** | **1.042** | **909** | **5.074** |

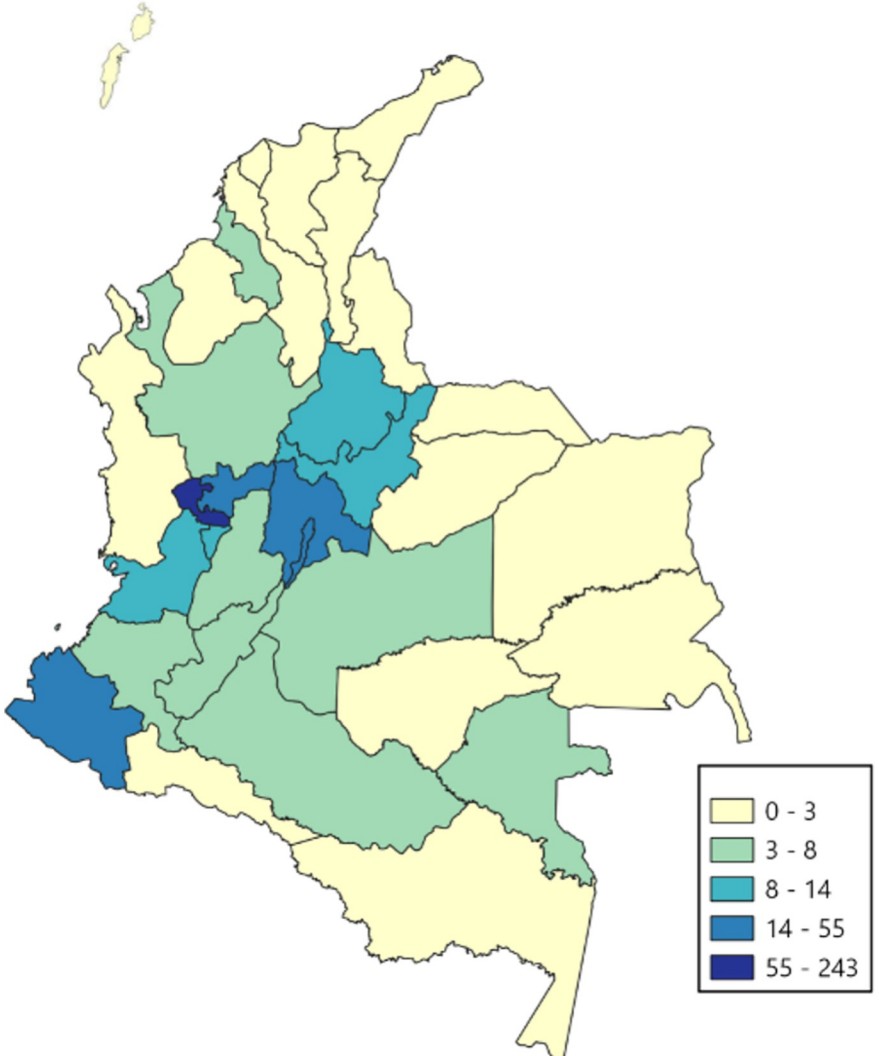

**Fig 2. Prevalence of sleep apnea in children and adolescents by department, Colombia, 2019.** "Elaborated by the authors. Map shapes/layers reprinted from https://www.datos.gov.co/Mapas-Nacionales/Departamentos-y-municipios-de-Colombia/xdk5-pm3f under a CC BY license, covered by Law 1712 of 2014 of the Colombian Ministry of Information and Communication Technologies, 2022".

undergone diagnostic testing confirming SA. Multiple strategies exist for pediatric SA diagnosis and validation and standardization is further required for both children and adult studies [32].

The findings of age-related higher prevalence of SA in the 6 to 11 years age group and in males comes in accordance with the described in other studies [33]. This has been reported in relation to hypertrophy of the tonsils and adenoid tissue which narrows the airway [34], this is also supported with our study's finding that 1737 children with SA in Colombia have been managed with ENT surgery in the five years. The approach to SA diagnosis and treatment goes beyond sleep studies and requires a multidisciplinary strategy [35]. Multiple management strategies exist for pediatric SA [9,12], some of them have important variations such as tonsillectomy versus tonsillotomy for oSDB [36]. Health registries such as SISPRO are limited in the

**Table 4. Diagnostic procedures performed in children and adolescents with sleep apnea, Colombia 2017–2021.**

| Diagnostic procedure | Age group | Year | | | | | |
|---|---|---|---|---|---|---|---|
| | | 2017 | 2018 | 2019 | 2020 | 2021 | Total |
| Polisomnography | < 1 | 0 | 0 | 0 | 0 | 6 | 6 |
| | 1 to 5 | 50 | 92 | 221 | 106 | 190 | 630 |
| | 6 to 11 | 308 | 266 | 431 | 139 | 261 | 1369 |
| | 12 to 17 | 199 | 154 | 234 | 85 | 195 | 840 |
| | Total | 557 | 512 | 886 | 330 | 652 | 2845 |
| Capnography | < 1 | 0 | 0 | 0 | 2 | 10 | 12 |
| | 1 to 5 | 3 | 24 | 49 | 24 | 39 | 136 |
| | 6 to 11 | 28 | 41 | 55 | 29 | 60 | 206 |
| | 12 to 17 | 6 | 18 | 28 | 18 | 28 | 95 |
| | Total | 37 | 83 | 132 | 73 | 137 | 449 |
| Nasolaryngoscopy | < 1 | 0 | 0 | 0 | 0 | 3 | 3 |
| | 1 to 5 | 1 | 12 | 68 | 78 | 68 | 224 |
| | 6 to 11 | 11 | 83 | 222 | 145 | 140 | 593 |
| | 12 to 17 | 5 | 68 | 142 | 101 | 96 | 405 |
| | Total | 17 | 163 | 432 | 324 | 307 | 1225 |
| Transthoracic ecocardiography | < 1 | 0 | 0 | 0 | 1 | 4 | 5 |
| | 1 to 5 | 4 | 20 | 27 | 18 | 26 | 93 |
| | 6 to 11 | 4 | 12 | 24 | 8 | 27 | 75 |
| | 12 to 17 | 1 | 5 | 11 | 9 | 14 | 40 |
| | Total | 9 | 37 | 62 | 36 | 71 | 213 |

information they provide regarding variations in diagnostic and therapeutic procedures such as the ones presented in our study.

The changes in prevalence through the five years analyzed are quite remarkable. Between 2017 and 2019 there was a clear tendency of increasing number of cases, diagnostic and therapeutic procedures; making 2019 the year with the highest and most reliable/closest to reality

**Table 5. Diagnostic procedures performed in children and adolescents with sleep apnea, Colombia 2017–2021.**

| Therapeutic procedures | Age group | Year | | | | | |
|---|---|---|---|---|---|---|---|
| | | 2017 | 2018 | 2019 | 2020 | 2021 | Total |
| PAP therapy | < 1 | 0 | 0 | 0 | 0 | 0 | 0 |
| | 1 to 5 | 0 | 0 | 1 | 0 | 2 | 2 |
| | 6 to 11 | 0 | 0 | 2 | 4 | 4 | 7 |
| | 12 to 17 | 2 | 1 | 21 | 20 | 18 | 32 |
| | Total | 2 | 1 | 24 | 24 | 24 | 41 |
| Amigdalectomy | < 1 | 0 | 0 | 0 | 0 | 0 | 0 |
| | 1 to 5 | 1 | 7 | 56 | 21 | 17 | 102 |
| | 6 to 11 | 62 | 194 | 247 | 79 | 28 | 610 |
| | 12 to 17 | 26 | 59 | 64 | 14 | 7 | 170 |
| | Total | 89 | 260 | 367 | 114 | 52 | 882 |
| Adenoidectomy | < 1 | 0 | 0 | 0 | 0 | 0 | 0 |
| | 1 to 5 | 3 | 10 | 55 | 22 | 17 | 107 |
| | 6 to 11 | 72 | 215 | 232 | 60 | 26 | 604 |
| | 12 to 17 | 24 | 54 | 47 | 11 | 8 | 144 |
| | Total | 99 | 279 | 334 | 93 | 51 | 855 |

measure of prevalence calculated to date (21.1 by 100000 inhabitants). Then, the number of cases significantly decreased in 2020 and 2021, this could be in evident relation to the COVID-19 pandemic effect on sleep medicine services worldwide [37]. Sleep clinics had to close temporarily to focus human and infrastructure resources in the pandemic. Also, the number of diagnostic procedures and particularly polysomnography diminished remarkably [37]. A communication was made by sleep specialists in Colombia to guide sleep medicine services nationwide during the pandemic [38].

The ample differences in prevalence between geographic regions could obey to difficulties in healthcare services access by patients. Some of the departments with the lowest prevalence have in common a low population density and less urban development. The graphic prevalence distribution could also show a concentration of sleep medicine services in principal and developed cities, which could hinder access to diagnosis of SA to patients residing far from city centers this could reflect on the importance of validating and including other strategies other than conventional polysomnography such as portable polysomnography [39,40], bio-impedance [41], heart rate variability monitoring [42] and peripheral arterial tonometry [43] as alternatives for children residing far from sleep medicine services. The fact that most of the patients belong to the contributory regime, may reflect inequity in access a sleep medicine services. This could lead to sub diagnosis in the most vulnerable, underserved part of the Colombian population leaving them at risk for the untreated SA health and quality of life consequences; this same pattern has been found in other SISPRO registry studies in Colombia [24,30,44]. Also, our findings suggest low usage of capnography in polysomnographic studies which is recommended as per American Academy of Sleep Medicine manuals [45].

Acknowledging the limitations of our study, it is important to clarify that this study provides an estimated and not a real prevalence of SA in Colombia since the study design was not developed as a census. This prevalence depends on adequate diagnosis and registry of information in RIPS by every healthcare provider in the country. SISPRO registers the principal and related diagnosis, SA could be selected by providers as a related diagnosis of some of the SA associated comorbidities such as obesity, adeno-tonsillar hypertrophy, Down syndrome, craniofacial abnormalities and cardiovascular disease. The measurement of the prevalence of SA in relation to some of the listed diseases could be a proposal for future studies with a similar methodology.

Another limitation inherent to the way that SISPRO manages ICD-10 diagnostic categories is that our study comprises both OSA and CSA which, differ in diagnostic criteria [1]. Nevertheless, reports have been consistent in the fact that the CSA only accounts for approximately 5% of all SA cases in children [46]. Also, SA diagnosis has evolved in time and many physicians worldwide currently use the 3$^{rd}$ version of the International Classification of Sleep Disorders (ICSD III) [1] and not the ICD-10 criteria, nevertheless, we consider this limitation unlikely to result in significant bias, since the patients diagnosed by ICSD criteria should have also been included in SISPRO by the healthcare providers using an ICD-10 code. Also, due to the nature of the data available in SISPRO, our study does not allow for evaluation of other important aspects of disease such as severity; given the fact that most of the children in the registry were not reported to have initiated PAP therapy or had surgery performed we suspect most cases would be mild or moderate in severity. For future prevalence estimations and to strengthen the current public health data bases, sleep medicine services should standardize and properly inform diagnostic categories to RIPS, ideally healthcare registries should allow for further differentiation between OSA and CSA. Further studies (e.g. multicenter nationwide studies) are required to best resolve these limitations inherent to the use of healthcare registries to address complex diseases sucha as sleep apnea.

## Conclusion

This study estimates the prevalence of SA in Colombia based on the healthcare system's registry during five years. When comparing it to other studies performed to date it suggests sub diagnosis and sub registry of SA nationwide. There was a decrease in SA diagnosis in 2020 and 2021 related to the COVID-19 pandemic. This data calls upon early detection, adequate evaluation and management of this disease that consumes a great quota of resources.

## Supporting information

**S1 Data.**
(XLSX)

## Author Contributions

**Conceptualization:** Alan Waich, Margarita Manrique Andrade, Julio Cesar Castellanos Ramírez, Liliana Otero Mendoza, Sonia Maria Restrepo Gualteros, Olga Patricia Panqueva, Patricia Hidalgo Martínez.

**Data curation:** Margarita Manrique Andrade, Julieth Andrea Castañeda Aza.

**Formal analysis:** Margarita Manrique Andrade, Julieth Andrea Castañeda Aza.

**Investigation:** Alan Waich, Juanita Ruiz Severiche, Margarita Manrique Andrade, Julieth Andrea Castañeda Aza, Julio Cesar Castellanos Ramírez, Sonia Maria Restrepo Gualteros, Olga Patricia Panqueva.

**Methodology:** Alan Waich, Juanita Ruiz Severiche, Margarita Manrique Andrade, Julieth Andrea Castañeda Aza, Liliana Otero Mendoza, Sonia Maria Restrepo Gualteros, Olga Patricia Panqueva.

**Project administration:** Alan Waich.

**Resources:** Liliana Otero Mendoza.

**Supervision:** Alan Waich.

**Visualization:** Alan Waich, Julieth Andrea Castañeda Aza.

**Writing – original draft:** Alan Waich, Juanita Ruiz Severiche, Sonia Maria Restrepo Gualteros.

**Writing – review & editing:** Alan Waich, Juanita Ruiz Severiche, Margarita Manrique Andrade, Julio Cesar Castellanos Ramírez, Liliana Otero Mendoza, Sonia Maria Restrepo Gualteros, Olga Patricia Panqueva, Patricia Hidalgo Martínez.

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
