## [Decision Letter · Decision Letter 0]

24 Jun 2022

PONE-D-22-09943Prevalence of sleep apnea in children and adolescents in Colombia according to the national health registry, 2017-2021PLOS ONE

Dear Dr. Waich,

Thank you for submitting your manuscript to PLOS ONE. After careful consideration, we feel that it has merit but does not fully meet PLOS ONE’s publication criteria as it currently stands. Therefore, we invite you to submit a revised version of the manuscript that addresses the points raised during the review process.

We look forward to receiving your revised manuscript.

Kind regards,

Kuo-Cherh Huang

Academic Editor

PLOS ONE

Journal Requirements:

5. We note that Figure 2 in your submission contain map images which may be copyrighted. All PLOS content is published under the Creative Commons Attribution License (CC BY 4.0), which means that the manuscript, images, and Supporting Information files will be freely available online, and any third party is permitted to access, download, copy, distribute, and use these materials in any way, even commercially, with proper attribution. For these reasons, we cannot publish previously copyrighted maps or satellite images created using proprietary data, such as Google software (Google Maps, Street View, and Earth). For more information, see our copyright guidelines: http://journals.plos.org/plosone/s/licenses-and-copyright.

Additional Editor Comments:

Dear Dr. Waich

We appreciate your submission to PLOS ONE. Both reviewers have provided a variety of important concerns and helpful suggestions. Please respond carefully to their suggestions. In particular, please pay attention to the critical point raised by Reviewer 2 as regards the diversity of methods used for the diagnosis of sleep apnea in your study.

Kuo-Cherh Huang

Reviewers' comments:

Reviewer's Responses to Questions

**Comments to the Author**

1. Is the manuscript technically sound, and do the data support the conclusions?

Reviewer #1: Yes

Reviewer #2: No

2. Has the statistical analysis been performed appropriately and rigorously? 

Reviewer #1: Yes

Reviewer #2: No

3. Have the authors made all data underlying the findings in their manuscript fully available?

Reviewer #1: Yes

Reviewer #2: Yes

4. Is the manuscript presented in an intelligible fashion and written in standard English?

Reviewer #1: Yes

Reviewer #2: Yes

5. Review Comments to the Author

Reviewer #1: Introduction

- line 57, Obstructive sleep-disordered breathing (oSDB) is a condition that encompasses breathing problems when asleep, due to an obstruction of the upper airways, ranging in severity from simple snoring to obstructive sleep apnoea syndrome (OSAS). It affects both children and adults. In children, hypertrophy of the tonsils and adenoid tissue is thought to be the commonest cause of oSDB. As such, tonsillectomy - with or without adenoidectomy - is considered an appropriate first-line treatment for most cases of paediatric oSDB. In otherwise healthy children, without a syndrome, of older age (five to nine years), and diagnosed with mild to moderate OSAS by PSG, there is moderate quality evidence that adenotonsillectomy provides benefit in terms of quality of life, symptoms and behaviour as rated by caregivers and high quality evidence that this procedure is beneficial in terms of PSG parameters. At the same time, high quality evidence indicates no benefit in terms of objective measures of attention and neurocognitive performance compared with watchful waiting. Furthermore, PSG recordings of almost half of the children managed non-surgically had normalised by seven months, indicating that physicians and parents should carefully weigh the benefits and risks of adenotonsillectomy against watchful waiting in these children. This is a condition that may recover spontaneously over time. For non-syndromic children classified as having oSDB on purely clinical grounds but with negative PSG recordings, the evidence on the effects of adenotonsillectomy is of very low quality and is inconclusive.Low-quality evidence suggests that adenotonsillectomy and CPAP may be equally effective in children with Down syndrome or MPS diagnosed with mild to moderate OSAS by PSG. please discuss and cite doi:10.1002/14651858.CD011165.pub2

- line 65, an interesting systematic review analyzed the correlation between changes in behavior and cognitive outcomes after AT were according to the scores post-AT in almost all studies. After comparing the AT group and control group, only one study had no difference that reached significance at one year post-AT. In another study, it did not show any significant improvement in terms of all behavioural and cognitive outcomes. The questionnaires on sleep-related quality of life after AT (PSQ-SRBD or ESS or OSA-18 or KOSA) may improve with positive changes in sleep parameters (AHI, ODI and SpO2). Furthermore, there is a significantly higher decrease in OSAS symptoms than the pre-AT baseline score. please discuss and cite doi:10.3390/children8100921

Methods

Please apply the latest strobe guidelines, consort model and equator.

Discussion

- To assess whether partial removal of the tonsils (intracapsular tonsillotomy) is as effective as total removal of the tonsils (extracapsular tonsillectomy) in relieving signs and symptoms of oSDB in children, and has lower postoperative morbidity and fewer complications. For children with oSDB selected for tonsil surgery, tonsillotomy probably results in a faster return to normal activity (four days) and in a slight reduction in postoperative complications requiring medical intervention in the first week after surgery. This should be balanced against the clinical effectiveness of one operation over the other. However, this is not possible to determine in this review as data on the long-term effects of the two operations on oSDB symptoms, quality of life, oSDB recurrence and need for reoperation are limited and the evidence is of very low quality leading to a high degree of uncertainty about the results. please discuss and cite doi:10.1002/14651858.CD011365.pub2

Reviewer #2: Dear author,

I appreciate the fact that your article first focused on the analysis of the prevalence in the pediatric population of sleep apnea in Colombia. It is definitely a good starting point for the study of this pathology now widely spread and of great interest to the general population. It is evident, when reading the article, the difficulties encountered in data collection and the diversity of methods used for the diagnosis of sleep apnea.

My advice is to prefer standard diagnostic tools for apnea such as polysomnography or pulse oximetry (preferred method in pediatric age). This would also allow a better differential diagnosis between central and obstructive apneas. You could try to conduct a nationwide multicenter study rather than the Colombian registry analysis to circumvent this problem. Therefore, you could use the data you collected on posing about the lack of standard methods for the study of apnea and not just subdiagnosis.

I wish you a good continuation of your work.

6. PLOS authors have the option to publish the peer review history of their article (what does this mean?). If published, this will include your full peer review and any attached files.

Reviewer #1: No

Reviewer #2: No

---

## [Author Response · Author response to Decision Letter 0]

29 Jul 2022

Dr. Kuo-Cherh Huang

Academic Editor

PLOS ONE

PONE-D-22-09943

Prevalence of sleep apnea in children and adolescents in Colombia according to the national health registry, 2017-2021

Dear Dr. Kuo-Cherh Huang

Academic Editor

PLOS ONE

We thank the reviewers and the journal’s editorial team for their valuable comments. Please find attached a revised version of the manuscript. Below you can find a point-by point response to the reviewers’ comments.

We look forward your response.

Kind regards, 

The authors

Journal Requirements:

RESPONSE: Thank you for the observation. We have reviewed the PLOS ONE style requirements. We have made corrections in file naming and in the manuscript following the style templates.

RESPONSE: We provided additional details regarding participant consent. We included a new Ethical considerations and Data availability statement sub heading in the revised manuscript as follows: The study protocol was reviewed and approved by the Research and Ethics Committee of Hospital Universitario San Ignacio and Pontificia Universidad Javeriana, both located in Bogota, Colombia. (FM-CIE-0473-21). The study was classified as no risk research and conducted in agreement with the Helsinki Declaration and Resolution 008430 of 1993 issued by the Colombian Ministry of Health. Data collected for analysis came from SISPRO, the Colombian national health registry [17]. Data is fully anonymized in the source, before being accessed by researchers. Thus, a waiver for informed consent was obtained. The raw data is available publicly or under request at https://www.sispro.gov.co

RESPONSE: Thank you for the comment. We have updated the Cover letter with a Data Availability Statement. Data collected for analysis came from SISPRO, the Colombian national health registry. Data in SISPRO is fully anonymized irreversibly before being accessed by researchers. The raw data is available publicly or under request at https://www.sispro.gov.co . We have attached a supplement with the data set underlying the results. We also included the SISPRO website within the manuscript in which interested researchers could gain access to the information.

RESPONSE: Thank you for the comment. We have included a full ethics statement in the Methods section of the revised manuscript file including the full name of ethics committee that approved the study and waived the requirement of informed consent since data is irreversibly anonymized at the primary source (SISPRO).

5. We note that Figure 2 in your submission contain map images which may be copyrighted. All PLOS content is published under the Creative Commons Attribution License (CC BY 4.0), which means that the manuscript, images, and Supporting Information files will be freely available online, and any third party is permitted to access, download, copy, distribute, and use these materials in any way, even commercially, with proper attribution. For these reasons, we cannot publish previously copyrighted maps or satellite images created using proprietary data, such as Google software (Google Maps, Street View, and Earth). For more information, see our copyright guidelines: http://journals.plos.org/plosone/s/licenses-and-copyright.

RESPONSE: Thank you for the comment. The figures were elaborated directly by the authors and are not subject to copyright. We used the software QGIS (2009) which is an open access, open code geographical data system. The shapes/layers used to elaborate the map come from a public domain, open access and free use data available at https://www.datos.gov.co/Mapas-Nacionales/Departamentos-y-municipios-de-Colombia/xdk5-pm3f

This public domain is covered by Law 1712 of 2014 of transparency and access to national public information of Colombia which states that citizens (e.g., the authors) can access this public data freely and without restrictions and can be used by third parties. We have updated the figure caption of the copyrighted figure as follows: “Elaborated by the authors. Map shapes/layers from https://www.datos.gov.co/Mapas-Nacionales/Departamentos-y-municipios-de-Colombia/xdk5-pm3f under a CC BY license, covered by Law 1712 of 2014 of the Colombian Ministry of Information and Communication Technologies, 2022.”

Additional Editor Comments:

Dear Dr. Waich

We appreciate your submission to PLOS ONE. Both reviewers have provided a variety of important concerns and helpful suggestions. Please respond carefully to their suggestions. In particular, please pay attention to the critical point raised by Reviewer 2 as regards the diversity of methods used for the diagnosis of sleep apnea in your study.

Kuo-Cherh Huang

Reviewers' comments:

Reviewer's Responses to Questions

Comments to the Author

1. Is the manuscript technically sound, and do the data support the conclusions?

Reviewer #1: Yes

Reviewer #2: No

2. Has the statistical analysis been performed appropriately and rigorously? 

Reviewer #1: Yes

Reviewer #2: No

3. Have the authors made all data underlying the findings in their manuscript fully available?

Reviewer #1: Yes

Reviewer #2: Yes

4. Is the manuscript presented in an intelligible fashion and written in standard English?

Reviewer #1: Yes

Reviewer #2: Yes

5. Review Comments to the Author

Reviewer #1: Introduction

- line 57, Obstructive sleep-disordered breathing (oSDB) is a condition that encompasses breathing problems when asleep, due to an obstruction of the upper airways, ranging in severity from simple snoring to obstructive sleep apnoea syndrome (OSAS). It affects both children and adults. In children, hypertrophy of the tonsils and adenoid tissue is thought to be the commonest cause of oSDB. As such, tonsillectomy - with or without adenoidectomy - is considered an appropriate first-line treatment for most cases of paediatric oSDB. In otherwise healthy children, without a syndrome, of older age (five to nine years), and diagnosed with mild to moderate OSAS by PSG, there is moderate quality evidence that adenotonsillectomy provides benefit in terms of quality of life, symptoms and behaviour as rated by caregivers and high quality evidence that this procedure is beneficial in terms of PSG parameters. At the same time, high quality evidence indicates no benefit in terms of objective measures of attention and neurocognitive performance compared with watchful waiting. Furthermore, PSG recordings of almost half of the children managed non-surgically had normalised by seven months, indicating that physicians and parents should carefully weigh the benefits and risks of adenotonsillectomy against watchful waiting in these children. This is a condition that may recover spontaneously over time. For non-syndromic children classified as having oSDB on purely clinical grounds but with negative PSG recordings, the evidence on the effects of adenotonsillectomy is of very low quality and is inconclusive.Low-quality evidence suggests that adenotonsillectomy and CPAP may be equally effective in children with Down syndrome or MPS diagnosed with mild to moderate OSAS by PSG. please discuss and cite doi:10.1002/14651858.CD011165.pub2

RESPONSE: Thank you for the comment, we have added additional data in the revised manuscript regarding oSDB and the evidence of the effect of adenotonsillectomy and other management options in PSG values and other aspects of disease in children. We have discussed and cited the recommended reference.

- line 65, an interesting systematic review analyzed the correlation between changes in behavior and cognitive outcomes after AT were according to the scores post-AT in almost all studies. After comparing the AT group and control group, only one study had no difference that reached significance at one year post-AT. In another study, it did not show any significant improvement in terms of all behavioural and cognitive outcomes. The questionnaires on sleep-related quality of life after AT (PSQ-SRBD or ESS or OSA-18 or KOSA) may improve with positive changes in sleep parameters (AHI, ODI and SpO2). Furthermore, there is a significantly higher decrease in OSAS symptoms than the pre-AT baseline score. please discuss and cite doi:10.3390/children8100921

RESPONSE: Thank you for the comment, we have discussed and cited the recommended reference. Also, we added a sentence regarding the evidence of adenotonsillectomy and other therapeutical procedures to treat OSA improving PSG values, symptoms and quality of life of children with OSA.

Methods

Please apply the latest strobe guidelines, consort model and equator.

RESPONSE: Thank you for the comment. Since our study is observational, we have applied the latest STROBE guidelines.

Discussion

- To assess whether partial removal of the tonsils (intracapsular tonsillotomy) is as effective as total removal of the tonsils (extracapsular tonsillectomy) in relieving signs and symptoms of oSDB in children, and has lower postoperative morbidity and fewer complications. For children with oSDB selected for tonsil surgery, tonsillotomy probably results in a faster return to normal activity (four days) and in a slight reduction in postoperative complications requiring medical intervention in the first week after surgery. This should be balanced against the clinical effectiveness of one operation over the other. However, this is not possible to determine in this review as data on the long-term effects of the two operations on oSDB symptoms, quality of life, oSDB recurrence and need for reoperation are limited and the evidence is of very low quality leading to a high degree of uncertainty about the results. please discuss and cite doi:10.1002/14651858.CD011365.pub2

RESPONSE: Thank you for the comment, we have discussed and cited the recommended reference. We added a sentence regarding the different therapeutic options existing for pediatric SA and that some have variations such as intracapsular vs. extracapsular adenotonsillectomy since these specific data does not tend to appear in health registries such as SISPRO.

Reviewer #2: Dear author,

I appreciate the fact that your article first focused on the analysis of the prevalence in the pediatric population of sleep apnea in Colombia. It is definitely a good starting point for the study of this pathology now widely spread and of great interest to the general population. It is evident, when reading the article, the difficulties encountered in data collection and the diversity of methods used for the diagnosis of sleep apnea.

My advice is to prefer standard diagnostic tools for apnea such as polysomnography or pulse oximetry (preferred method in pediatric age). This would also allow a better differential diagnosis between central and obstructive apneas. You could try to conduct a nationwide multicenter study rather than the Colombian registry analysis to circumvent this problem. Therefore, you could use the data you collected on posing about the lack of standard methods for the study of apnea and not just subdiagnosis. I wish you a good continuation of your work.

RESPONSE: Thank you for the comment and review. We added new sentences throughout the revised manuscript regarding the diversity of diagnostic methods for SA. Since the patients in our study were given a confirmed principal diagnosis of SA by a healthcare professional, they should have undergone polysomnographic testing confirming SA (central and/or obstructive). 

We agree with the reviewer on the limitations that are inherent to the use of a healthcare registries methodology to collect data and that are reflected in the discussion section of the manuscript. We agree with the reviewer in that a nationwide multicenter study could allow for a better differential diagnosis between central and obstructive apneas, our study would also invite healthcare authorities to start discriminating between central and obstructive sleep apneas in the national health registry. We expect this first analysis to invite researchers to design and perform studies with different methodologies such as a nationwide multicenter study proposed by the reviewer, we have added further insight in this matter in the discussion section of the revised manuscript. 

In the future, a prospective study is proposed to include patients diagnosed with sleep apnea using PSG as a diagnostic method to achieve greater precision between central and obstructive sleep apnea.

6. PLOS authors have the option to publish the peer review history of their article (what does this mean?). If published, this will include your full peer review and any attached files.

Do you want your identity to be public for this peer review? For information about this choice, including consent withdrawal, please see our Privacy Policy.

Reviewer #1: No

Reviewer #2: No

RESPONSE: Thank you for the comment, we have uploaded the updated figure files to PACE digital diagnostic tool.

---

## [Decision Letter · Decision Letter 1]

8 Aug 2022

Prevalence of sleep apnea in children and adolescents in Colombia according to the national health registry, 2017-2021

PONE-D-22-09943R1

Dear Dr. Waich,

We’re pleased to inform you that your manuscript has been judged scientifically suitable for publication and will be formally accepted for publication once it meets all outstanding technical requirements.

Kind regards,

Kuo-Cherh Huang

Academic Editor

PLOS ONE

Additional Editor Comments (optional):

Reviewers' comments:

Reviewer's Responses to Questions

**Comments to the Author**

1. If the authors have adequately addressed your comments raised in a previous round of review and you feel that this manuscript is now acceptable for publication, you may indicate that here to bypass the “Comments to the Author” section, enter your conflict of interest statement in the “Confidential to Editor” section, and submit your "Accept" recommendation.

Reviewer #1: All comments have been addressed

2. Is the manuscript technically sound, and do the data support the conclusions?

Reviewer #1: (No Response)

3. Has the statistical analysis been performed appropriately and rigorously? 

Reviewer #1: (No Response)

4. Have the authors made all data underlying the findings in their manuscript fully available?

Reviewer #1: (No Response)

5. Is the manuscript presented in an intelligible fashion and written in standard English?

Reviewer #1: (No Response)

6. Review Comments to the Author

Reviewer #1: Dear author the paper is improved and could be accepted. Well done. Really interesting, will add several informations to the literature

7. PLOS authors have the option to publish the peer review history of their article (what does this mean?). If published, this will include your full peer review and any attached files.

Reviewer #1: No

---

## [Editor Report · Acceptance letter]

22 Aug 2022

PONE-D-22-09943R1 

Prevalence of sleep apnea in children and adolescents in Colombia according to the national health registry 2017-2021 

Dear Dr. Waich:

I'm pleased to inform you that your manuscript has been deemed suitable for publication in PLOS ONE. Congratulations! Your manuscript is now with our production department. 

Kind regards, 

on behalf of

Dr. Kuo-Cherh Huang 

Academic Editor

PLOS ONE